# Enhanced formation of methane hydrate from active ice with high gas uptake

Peng Xiao[1], Juan-Juan Li[1], Wan Chen[1], Wei-Xin Pang[2], Xiao-Wan Peng[1], Yan Xie[1], Xiao-Hui Wang [1], Chun Deng[1], Chang-Yu Sun [1] ✉, Bei Liu[1] ✉, Yu-Jie Zhu[1], Yun-Lei Peng[1], Praveen Linga [3] ✉ & Guang-Jin Chen [1] ✉

Gas hydrates provide alternative solutions for gas storage & transportation and gas separation. However, slow formation rate of clathrate hydrate has hindered their commercial development. Here we report a form of porous ice containing an unfrozen solution layer of sodium dodecyl sulfate, here named active ice, which can significantly accelerate gas hydrate formation while generating little heat. It can be readily produced via forming gas hydrates with water containing very low dosage (0.06 wt% or 600 ppm) of surfactant like sodium dodecyl sulfate and dissociating it below the ice point, or by simply mixing ice powder or natural snow with the surfactant. We prove that the active ice can rapidly store gas with high storage capacity up to 185 $V_g V_w^{-1}$ with heat release of ~18 kJ mol$^{-1}$ CH$_4$ and the active ice can be easily regenerated by depressurization below the ice point. The active ice undergoes cyclic ice−hydrate−ice phase changes during gas uptake/release, thus removing most critical drawbacks of hydrate-based technologies. Our work provides a green and economic approach to gas storage and gas separation and paves the way to industrial application of hydrate-based technologies.

Gas hydrates have attracted much attention because not only natural gas hydrates are believed to be the largest natural gas resource on Earth, but also gas hydrates provide alternative and innovative solutions for industrial requirements, e.g., natural gas storage & transportation and gas mixture separation. Gas hydrate forms when water molecules and gas molecules contact at low temperatures and high pressures; as a result, gas molecules are enwrapped in lattices that consisted of hydrogen−bonded water molecules[1]. Ideally, each unit volume of hydrate can store 180 volumes of natural gas[2]; different gases have different equilibrium conditions in hydrate formation; in addition, gas hydrate can be preserved under atmospheric pressure at 268.15 K[3]. These features make it a promising medium for gas storage and separation. However, the feasibility of these hydrate−based technologies is heavily reliant on the formation rate of gas hydrate, which is naturally slow.

Gas hydrate formation is an interfacial process between gas and water. For the hydrate formation without human intervention, a solid hydrate film with low permeability would spread over and seal off the interface, which hinders the contact of gas and water for continued gas hydrate formation. Consequently, a spontaneous hydrate growth always stagnates at film thickness less than 100 μm[4,5], leaving a large amount of unconverted water beneath the hydrate film. Besides that, the heat released during hydrate formation due to its exothermic nature also would retard hydrate formation in turn by rising the reactants' temperature. Thus, the slow hydrate formation results from the retarded mass transfer and the heat generation.

Efforts have been dedicated to remove or minimize mass transfer resistance and to accelerate hydrate formation[6–15]. Among them, using kinetic promoters like surfactants, e.g., sodium dodecyl sulfate (SDS), is among the most effective ways known today to accelerate hydrate

[1] State Key Laboratory of Heavy Oil Processing, China University of Petroleum, Beijing 102249, P. R. China. [2]State Key Laboratory of Natural Gas Hydrate, CNOOC Research Institute Co., Ltd., Beijing 100027, P. R. China. [3]Department of Chemical and Biomolecular Engineering, National University of Singapore, Singapore 117585, Singapore. ✉e-mail: cysun@cup.edu.cn; liub@cup.edu.cn; chepl@nus.edu.sg; gjchen@cup.edu.cn

formation. Another class of kinetic promoters that have gained recent prominence are amino acids[16]. The fastest hydrate formation in the presence of kinetic promoters reported so far could be accomplished at 11 min with water conversion over 80%[6]. Despite the conspicuous performance of kinetic promoters accelerating hydrate formation, scaling-up the hydrate production via using kinetic promoters may encounter three major challenges.

The accelerated hydrate formation comes from an upward and wall-adhering growth pattern of gas hydrate induced by kinetic promoters, which is known as wall−climbing effect[17]. This effect accelerates hydrate formation by sucking the aqueous solution onto the reactor wall and providing larger contact area for hydrate formation. Due to such hydrate growth pattern, the hydrate formation rate is closely related to the solution load in reactors; with the same reactor configuration and under similar thermodynamic conditions, overall hydrate formation rate decreases with the increase of solution load[18–20], which is unfavorable for scaling-up of the process. The second challenge hindering the scaling-up is the temperature increase due to the exothermic nature of the formation process; because of significant heat released during hydrate formation from water or aqueous solution (e.g., -9.07 kJ mol$^{-1}$ water or 54.44 kJ mol$^{-1}$ for methane hydrate[21]), the temperature of the liquids could rise by 4 ~ 8 K in the 0.5 - 2.7 h of formation processes[14,18,20]; a faster hydrate formation process required by industrial hydrate production would further increase the temperature, and then hamper the hydrate formation in turn. The third challenge is the inadequate utilization of the reactor space; plenty of gas space is required for the upward growth to avoid the blockage of the gas inlet/outlet; for that reason, gas hydrate formation in the presence of surfactants is generally conducted with gas volume of 65% ~ 90% in the reactors. Besides the three major challenges, the hollow appearance of the hydrate[22] formed with the wall−climbing effect could decrease the apparent gas storage density; meanwhile,

the serious bubble and foam generation during hydrate dissociation is also detrimental to industrial gas hydrate production.

In order to break through the limits of mass and heat transfer on hydrate formation rate and reduce the subsequent problems, here, we report an active ice approach and demonstrate its excellence as a formation medium for gas hydrate. We first use a kind of active ice produced by decomposing primary gas hydrate that is formed from SDS solution to form methane hydrate, and the hydrate formation typically completes within 5 min with gas uptake over than 170 $V_g V_w^{-1}$ at 272.65 K. Beside the rapid formation rate, the active ice exhibits multiple advantages, i.e., good repeatability, recyclability, compressibility, simple storage condition and low heat generation. Characterization results show that the active ice is porous and contains unfrozen SDS solution layer. Gas hydrate formation in this solution layer at temperature slightly lower than the ice point spawns a virtuous circle of ice melt−hydrate formation−ice melt, which hastens gas hydrate formation strikingly. The ice with unfrozen surfactant solution layer seems to be activated for gas hydrate, it is therefore an active ice. Inspired by this, we improve the preparation process of active ice without impairing its performance in gas hydrate formation. With the multiple advantages, the active ice approach is capable of overcoming the obstacles to scaling−up gas hydrate formation process for industrial purpose.

## Results

Being attracted by the porous morphologies of gas hydrate formed from aqueous SDS solution (CH$_4$−SDS hydrate), we wonder whether cage structures retain in the solid medium produced after hydrate dissociation below ice point and try to use it for reformation of gas hydrate. An active ice was made by decomposing a primary gas hydrate mass below the ice point, which was previously synthesized with methane and 600−ppm SDS solution (Fig. 1a−c. Supplementary Fig. 1), then methane hydrate was formed in it (Fig. 1d). When methane

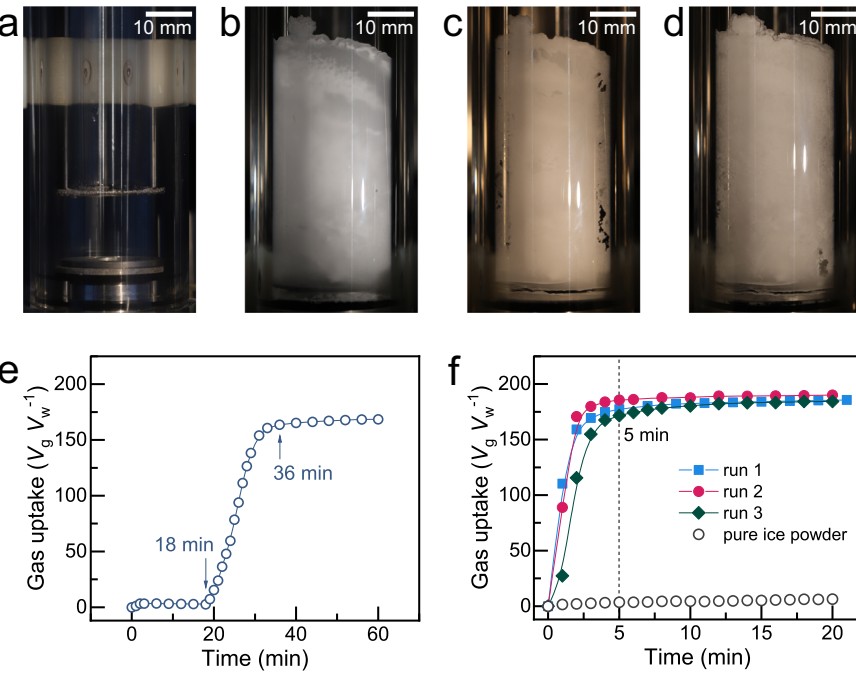

**Fig. 1 | Morphology change and kinetics of methane hydrate formation in fresh SDS solution and in active ice. a** fresh 600-ppm SDS aqueous solution. **b** Primary methane hydrate formed in the SDS solution at 277.15 K and initial pressure of 6.0 MPa. **c** Medium (so-called active ice) originated via dissociating hydrate shown in (**b**) at 272.65 K and atmospheric pressure. **d** Methane hydrate re-formed with the active ice shown in (**c**) at 272.65 K and initial pressure of 6.0 MPa; no wall−climbing

effect was observed during the hydrate formation process. **e** Gas uptake of methane hydrate formed in 600-ppm SDS solution at 272.65 K and initial pressure of 6.0 MPa, showing an induction time of 18 min. **f** Comparison of methane hydrate formation in the active ice and in pure ice powder at 272.65 K and initial pressure of 6.0 MPa; runs 1 - 3 were three independent experiments performed with the active ice; the particle size of the pure ice powder was 180 - 250 µm.

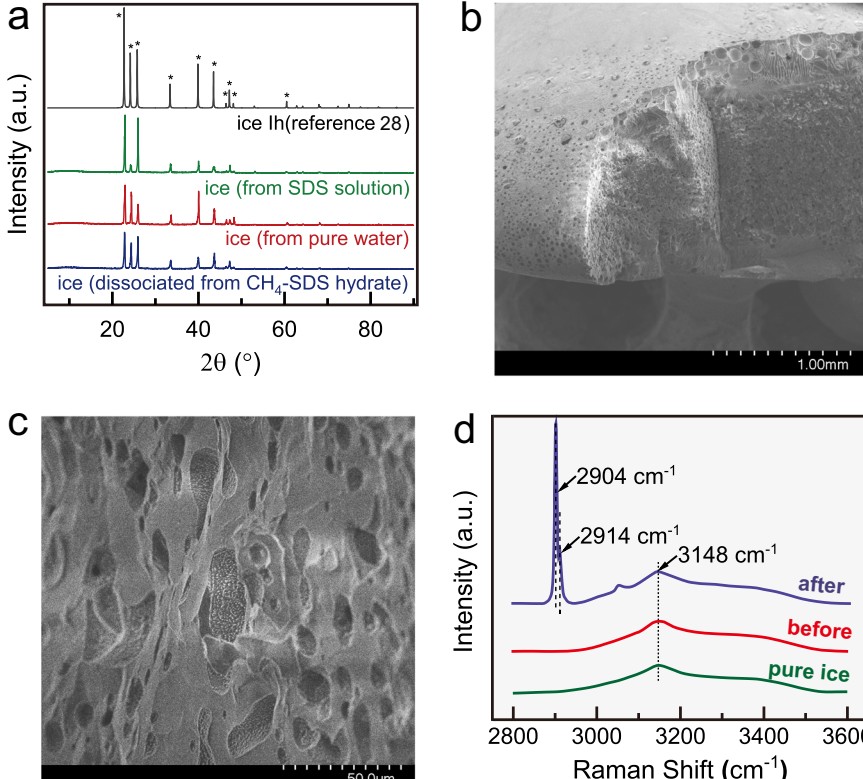

**Fig. 2 | Characterization of the active ice. a** PXRD patterns for the activate ice (dissociated from $CH_4$ − SDS hydrate), aqueous ice[28], and the ice of 600-ppm SDS solution. **b** SEM image of the active ice. **c** Pores on the activate ice. **d** Raman spectra for pure ice and the active ice before and after methane uptake; the peaks at 2904 $cm^{-1}$ and 2914 $cm^{-1}$ indicate the C − H stretching vibration of methane molecule in large and small cages of structure I methane hydrate, respectively[29]; 3000 - 3600 $cm^{-1}$ corresponds to O-H stretching; the peak at 3053 $cm^{-1}$ is the distinction between C-H stretching in methane hydrate and ice[30].

hydrate formed in 600−ppm SDS solution at 272.65 K and initial pressure of 6.0 MPa (Supplementary Fig. 2), the induction time for hydrate nucleation ranged from 5 to 82 min though stirring was adopted to accelerate it, and the time required to finish the hydrate formation after the induction period ranged from 18 to 32 min. In one of these formation experiments (Fig. 1e), the methane uptake of hydrate was 168.60 $V_g V_w^{-1}$ (volume of STP gas per volume of water; the theoretical value without the effect of thermodynamic promoters is 216 $V_g V_w^{-1}$ for structure I methane hydrate) after a 1 h formation process, including an induction time of 18 min. By contrast, the active ice gulps methane very rapidly without retard at 272.65 K (Fig. 1f). The equilibrium methane uptake reaches in 5 min and is encouragingly high as 185.70 $V_g V_w^{-1}$ or so; meanwhile, the wall−climbing effect[17], which always occurs when gas hydrate forms in surfactant solutions and potentially hinders the operation in hydrate production, could be eliminated in the active ice. This rapid hydrate formation implied that the major obstacle in hydrate formation, i.e., blocked mass transfer, could be overcome by the active ice approach.

Following PXRD measurements indicate this kind medium is normal ice still (Fig. 2a) although SEM images show that its porous characteristic is significant (Fig. 2b, c. Supplementary Fig. 3); the thermal property of the medium also support this inference (Supplementary Table 1). Considering pure ice powder does not exhibit such perfect gas uptake performance (Fig. 1f), we call this kind of medium active ice. The Raman spectrum measurements (Fig. 2d) suggest that the active ice and pure ice have the same Raman spectrum; after methane uptake in the active ice, phase transition of ice to clathrate hydrate has occurred, indicating that the gas uptake process in the active ice is actually a clathrate hydrate formation process, although little change in the morphology of the active ice could be observed by naked eye (refer to Fig. 1c, d). The characteristic that gas uptake occurs

only when pressure is higher than equilibrium pressure of hydrate formation (Supplementary Fig. 4) also supports the formation of clathrate hydrate.

Figure 3 shows the mechanism of fast hydrate formation in the active ice. Adsorbed SDS molecules on the surface of ice particles work like salt to decrease the local melting point and result in the occurrence of aqueous solution layer (Supplementary Table 2) on the surface when the temperature is set to 272.65 K or so, which is very close to the melting point of pure ice. Because of the promotion effect of SDS, this liquid layer forms hydrate promptly when the active ice contact to gas like methane under enough high pressure. The difference between the hydrate formation heat from liquid water (-9.07 kJ $mol^{-1}$ water[21]) and the ice melting heat (5.99 kJ $mol^{-1}$ water, Supplementary Table 1) guarantees the continuous melt of ice and generation of new liquid layer between the hydrate shell and residual ice core; the liquid layer then adsorbs SDS molecules, which are excluded in the pre−hydrate formation, to its surface; porous morphology of hydrate shell formed in the presence of SDS helps gas molecules migrate to the surface of the newly generated liquid layer rapidly and cause new hydrate formation. By this way, a virtuous circle of ice melt−hydrate formation−ice melt is established, which results in the rapid hydrate formation from the surface to center of active ice particles. This kind of melt-catalytic mechanism as we see it is supported by two facts. The first one is that the temperature increases from 272.65 K to higher than 278.0 K, which is higher than the melting point of ice, during hydrate formation when initial temperature is set to 272.65 K (Supplementary Fig. 5); this indicates that ice melting is thermodynamically possible. The second one is that both the hydrate formation rate and the maximum gas uptake are very low when temperature is set far lower than the ice point, e.g., ≤270.65 K (Supplementary Fig. 6); this result could be attributed to the impossibility to

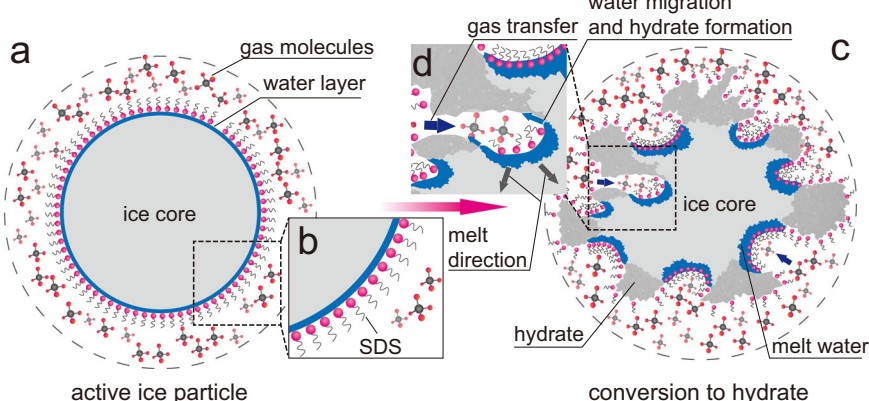

**Fig. 3 | Schematic illustration of gas hydrate formation on the active ice particle. a** Active ice particle before hydrate formation. **b** unfrozen aqueous solution layer on the particle because of the existence of SDS molecules. **c** gas hydrate formation on the active ice particle, causing irregular appearance change while leaving passages for gas transfer. **d** exothermic hydrate formation from aqueous solution layer and endothermic ice melt constitute the basis for rapid hydrate formation in the active ice.

form enough amount of liquid layer before and during hydrate formation under such lower temperature. Because of the porous or powdery characteristic of the active ice, gas could contact to all ice particles at the same time and forms hydrate uniformly, that is why we see a rapid gas uptake.

Since the surfactant played a critical role in the rapid gas hydrate formation in the active ice, we have further tested six different kinetic promoters to prepare the active ice to further extend the applicability to different surfactants, whose methane uptake profiles are shown in Supplementary Fig. 7. All these kinetic promoters display positive effects on accelerating hydrate formation while sodium dodecanoate and sodium dodecyl benzene sulfonate perform better than the rest. The comparison between Supplementary Figs. 7 and 8 indicate that the performance of a kinetic promoter in promoting gas hydrate formation in the active ice is closely related to its ability in promoting gas hydrate formation in aqueous solution; namely, if a promoter can significantly enhance gas hydrate formation in aqueous solution, it is also a potential additive for producing the active ice.

More tests have been performed to show the other feasibility and advantages for active ice to be used for gas storage or gas mixture separation. For example, active ice could be recycled (Fig. 4a, Supplementary Fig. 9) and be stored for long time (Fig. 4b), without losing its gas uptake performance; the optimized SDS dose in active ice is as low as $600 \, \text{mg L}^{-1}$ (Supplementary Fig. 10). Actually, the regeneration of the active ice does not refer to strictly reinstating the microstructure of the active ice, it only means recovering the ability of providing rapid gas uptake as evidenced in this work. In addition, high hydrate formation rate and gas uptake could be achieved under smaller driving force (the difference between actual hydrate formation pressure and equilibrium hydrate formation pressure). As shown in Fig. 4c, the cumulative gas uptake could reach $174.8 \, V_g \, V_w^{-1}$ within 5 min under 272.65 K and constant pressure of $3.0 \pm 0.02 \, \text{MPa}$, corresponding to a drive force of only 0.69 MPa, which is much smaller than that typically adopted in cases where aqueous SDS solution were used to form hydrate[18,23]. Surely, moderate operation pressure helps to reduce the operating cost of hydrate production. Moreover, another bottleneck besides the slow formation rate of the hydrate-based technologies, i.e., high formation heat, could be basically eliminated. Compared to the formation heat of higher than $50 \, \text{kJ mol}^{-1}$ gas for methane hydrate formed from water, the formation heat of methane in active ice is only $18.13 \, \text{kJ mol}^{-1} \, CH_4$[24]. The smaller heat release could bring about lower increment in temperature peak when the mass of active ice is multiplied in fast methane uptake. When the mass of active ice was increased from 5.0 to 79.0 g, the temperature peak only increased from 278.76 to 281.53 K in the fast gas uptake period (<5 min); meanwhile, the gas uptakes ($181.10 \, V_g \, V_w^{-1}$ and $173.21 \, V_g \, V_w^{-1}$) reached 91.19% and 90.97% of their final values, respectively (Supplementary Fig. 5c). In these experiments, methane hydrate formation in the active ice were accomplished within 5 min (gas uptake > 89% of its final value); this indicated that hydrate formation started simultaneously at multi points in the active ice, which is marginally affected by the amount of the active ice; all these results suggested an excellent potential of active ice in scaling-up the gas hydrate production. Apart from the formation heat, the effect of the packing bed's size on the heat removal plays an important role in the inevitable increase in temperature peak when the amount of active ice was multiplied; however, this issue could be solved by optimizing the heat exchange in the packing bed of active ice.

Although the porous or powdery morphology of active ice brings high gas uptake rate, it makes the apparent specific volume of active ice packing bed much bigger than that of ice crystal and results in lower apparent storage capacity. For example, the apparent storage capacity is only $83.92 \pm 2.6 \, V_g \, V_{bed}^{-1}$ (standard volume of gas per volume of active ice bed; free gas in the bed is included) if directly using the hollow active ice column produced by dissociating $CH_4 - SDS$ hydrate displaying in Fig. 1c. It is too low for industrial application. Hence compression is required to decrease the apparent volume and increase apparent gas storage capacity of active ice packing bed. Certainly, the compression will reduce both uptake rate and final gas uptake as shown in Fig. 4d. Fortunately, these negative effects are negligible under moderate compression, e.g., in cases when loosening coefficient $\alpha$, the ratio of the specific volume of active ice packing bed to the specific volume of perfect ice crystal of the same mass, retains higher than 1.398. Therefore, decreasing the apparent volume of the active ice (or gas hydrate) by compression and good regeneration capacity can be realized at the same time. However, this only happens at moderate compression levels. The apparent methane storage capacity of active ice packing bed reached $126.36 \, V_g \, V_{bed}^{-1}$ under the optimized loosening coefficient $\alpha = 1.492$ and 5.76 MPa (residual pressure after hydrate formation, Supplementary Fig. 11), which is 2.44 times higher than that of compressed natural gas under the same temperature and pressure. It is also much higher than that of dry ZIF-8 (a well-known metal-organic framework adsorbent) packing bed under similar thermodynamic conditions ($105.37 \, V_g \, V_{bed}^{-1}$ at 269.15 K and 5.25 MPa)[25].

Though the active ice has shown excellent performance in accelerating gas hydrate formation, the complicated preparation of it would

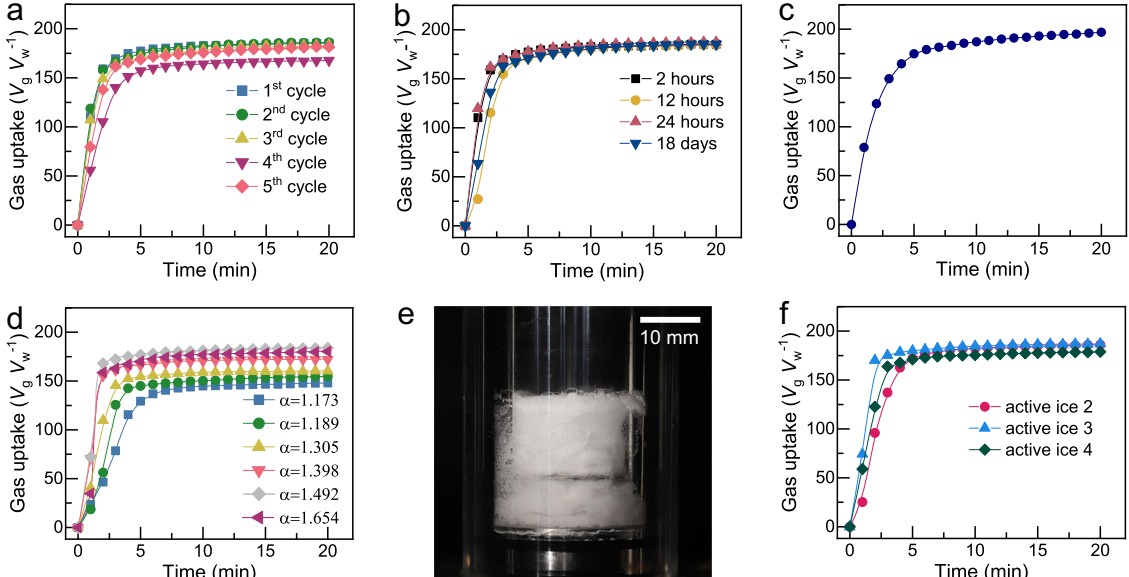

**Fig. 4 | Properties of the active ice in gas hydrate formation. a** Cumulative gas uptake of methane hydrate formed in a cycle test for the active ice. Methane hydrate was formed at 272.65 K and initial pressure of 6.0 MPa; the hydrate dissociation to active ice was conducted at 272.65 K and atmospheric pressure. **b** Cumulative gas uptake in a preservation test for the active ice; the active ice was preserved at 268.15 K for varying durations and then used for methane hydrate formation at 272.65 K and initial pressure of 6.0 MPa. **c** Cumulative gas uptake of methane hydrate formed in the active ice at constant pressure of 3.0 MPa and 272.65 K. **d** Cumulative gas uptake of methane hydrate formed in active ices that

compressed to different loosening coefficients; the temperature and the initial pressure for hydrate formation were 272.65 K and 6.0 MPa, respectively. **e** Photo of the methane hydrate formed in the compressed active ice with loosening coefficient of 1.55, showing no wall−climbing effect during formation. **f** Cumulative methane uptake of another three activate ices at 272.65 K and initial pressure of 6.0 MPa. Active ice 2: the ice dissociated from $CO_2$−SDS hydrate; active ice 3: ice powder of 600-ppm SDS solution, with particle size of 180 - 250 μm; active ice 4: natural snow mixed with 0.1 wt% SDS.

restrict its practical application. We have further attempted to simplify its preparation process based on the mechanism of fast gas hydrate formation in active ice. Those experiments show that the active ice prepared via dissociating $CO_2$−SDS hydrate below the ice point (active ice 2 in Fig. 4f) also works well, just like that originated from $CH_4$−SDS hydrate. This finding is of significance because it makes the manufacture of the active ice safer and more economic since $CO_2$ is incombustible and can form hydrate under milder conditions. It is more interesting to observe that the active ice prepared by grinding frozen aqueous SDS solution or by simply mixing natural snow with SDS also exhibits the same perfect gas uptake behavior (active ice 3 and 4, respectively, in Fig. 4f).

Finally, we demonstrate the use of active ice for the separation of several representative binary gas mixtures, $CH_4/CO_2$, $CH_4/H_2$, and $CH_4/C_2H_6$, and compared its separation performance with that of aqueous SDS solution. The results show that, compared with SDS solution, the active ice not only eliminated the induction time and diminished the time required for separating the gas mixtures, it also remarkably increased the separation factor compared to SDS solution (Supplementary Fig. 12 and Supplementary Table 3). For $CH_4/H_2$, a separation factor of 355.66 and a methane recovery of 38.22% could be acquired within 11 min with the active ice, which was much faster and more effective than that with SDS solution under similar conditions. With an obvious improvement in separation kinetics and better separation results, the feasibility of hydrate-based gas separation could, therefore, be improved with the active ice.

We propose a conceptual process with respect to the potential application of the active ice in natural gas storage and transportation. The active ice could be prepared by decomposing gas-promoter hydrate below the ice point, or grinding the ice of suitable kinetic promoters' aqueous solution, or just simply mixing ice powder or natural snow with kinetic promoters. As regards the most economical

or feasible preparation method, it depends on specific situations. After being produced and shaped in a factory, the active ice could be loaded in a tank on vehicles, in which built-in heat exchangers should be installed. The on-board tanks will be used for gas storage in the gas fields. Natural gas is precooled and injected into the tank and is expected to form gas hydrate rapidly with the active ice. The formed hydrate then will be degassed to atmospheric pressure and transported to its destination and natural gas is recovered along with the active ice. This kind of charging−discharging mode based on the rapid hydrate formation is similar to compressed natural gas (CNG) method; however, it is superior in the economic benefits, safety benefits and environmental friendliness perspective according to the literature[6]. As for the application of the active ice in gas mixture separation, traditional pressure swing adsorption (PSA) mode could be adopted because of the smaller heat release[26]. Because the uncertainty of the scaling-up effect, the above methods are only suggestions for the potential use of the active ice in gas storage and transportation and gas separation, and they must be developed and analyzed in further studies.

## Discussion
In summary, we have discovered a kind of active ice, which is of porous or powdery appearance and contains low-dose kinetic promoters. The promoters locally decrease the freezing point of water while keeping its effect on hydrate formation, thus create extensive active sites in the active ice at temperature slightly lower than the ice point of pure water. The spiral development of ice melting−hydrate formation−ice melting at the active sites enables minute-level and high-conversion gas hydrate formation even at very small driving force. The fast hydrate formation, in association with the small formation heat, the shapable packing bed of active ice, and the recyclability of the active ice, could overcome the main obstacles in realizing hydrate-based technologies. What's more, the greenness and cheapness of the active

ice, make it an excellent alternative sorbent for more extensive gas storage or separation.

## Methods

### Materials

Methane, carbon dioxide, hydrogen, and ethane with purity of 99.99% were purchased from Beijing AP BAIF Gas Industry CO., Ltd. Sodium dodecyl sulfate (SDS) with ACS reagent grade (>99%) was provided by Sigma-Aldrich. Sodium oleate with purity >97% and N-Carbobenzoxy-DL-leucine with purity >98% were purchased from Shanghai Aladdin Biochemical Technology Co., Ltd. Dodecylbenzenesulfonic acid (> 90%), Sodium dodecyl benzene sulfonate (>95%), Sodium laurylsulfonate (>98%) were provided by Shanghai Makclin Biochemical Co., Ltd. Deionized water was produced in our laboratory.

### Experimental setup

Supplementary Fig. 13 shows the schematic of the apparatus employed to produce the active ice and to evaluate its performance in hydrate formation. Briefly, a high-pressure sapphire cell with internal diameter of 2.54 cm and internal volume of 61.90 cm³ was used as the reactor. A piston was mounted at the bottom of the reactor to compress the active ice. An iron ring reciprocates inside the reactor and acts as a stirrer before gas hydrate formation in aqueous solution, which was driven by a group of magnets beside the reactor. A stainless-steel gas reservoir with an internal volume of 130.23 cm³ was used to precool gas. The reactor and the gas reservoir were mounted in an air bath (temperature stability = ±0.1 K). Two pressure transducers (0 ~ 20 MPa, ±0.1%FS; 0 ~ 10 MPa, ±0.1% FS) were used to measure the pressures of the gas reservoir and the reactor, respectively. The pressure data were logged by a data acquisition system for every 1 min. A buffer tank was used to precool the pressurizing fluid of the pump, and the inner volume of the buffer tank is 80.10 cm³. A stainless-steel reactor with internal diameter of 5.07 cm and 220.00 cm³ was used to conduct the scale-up hydrate formation experiments in Supplementary Fig. 5c, d; meanwhile, another gas reservoir with internal volume of 750.09 cm³ was used to precool the gas.

Another reactor was used coupled with a Raman spectrometer for in situ characterization of the active ice. The schematic of the reactor is shown in Supplementary Fig. 14. Two opposite sapphire windows were designed on the reactor for optical observation and Raman spectroscopy determination. Between the sapphire windows, a chamber with internal volume of approximately 1.4 cm³ is surrounded by the windows and the stainless-steel wall, and the in situ hydrate formation and dissociation for Raman characterization were performed in the chamber. An annular passage surrounding the chamber was designed as the flow channel for coolant to regulate the temperature in the chamber. A pt100 temperature sensor with accuracy of 0.05 K was inserted into the chamber. A pressure transducer with accuracy of 0.02 MPa was used to measure the pressure in the reactor.

### Preparation of the active ice

(1) methane hydrate formation-dissociation route. The active ice was prepared following a hydrate formation-dissociation method shown in Supplementary Fig. 1. A certain amount of SDS was dissolved in water first. Then, 10 g of the SDS solution was added into the sapphire reactor shown in Supplementary Fig. 13, and the reactor was sealed tight. The air in the reactor and the gas reservoir was purged by charging methane (primary gas) and vacuuming for three times. After that, the primary gas was charged into the gas reservoir to higher than 10 MPa. Subsequently, the temperature of the air bath was set to 277.15 K for hydrate formation. 3 h later, the temperature of the SDS solution and the primary gas reached 277.15 K after a long enough cooling. Then, the reactor was pressurized with the primary gas from the gas reservoir to 6.0 MPa, and the stirring device was

switched on to accelerate hydrate nucleation. When hydrate crystals appeared in the reactor, the stirring device was switched off to let hydrate grow quiescently. This hybrid approach enables the control of stochasticity associated with the nucleation of hydrate formation[27]. About 3 h later, the pressure drop caused by hydrate formation decreased to less than 20 kPa h⁻¹. Then, the formation of the hydrate (primary hydrate) is considered to be accomplished. The active ice was produced by dissociating the primary hydrate below the ice point. Before the hydrate dissociation, the temperature of the air bath was reset to 272.65 K. 3 h later, the primary hydrate was depressurized to atmospheric pressure within 20 s. Then, the primary hydrate dissociated at 272.65 K and under atmospheric pressure. The temperature and pressure change in the preparation of the active ice via dissociating methane hydrate are shown in Supplementary Fig. 15. The dissociation procedure lasted at least for 3 h to release most of the primary gas that stored in the primary hydrate. The ice-like substance left in the reactor after hydrate dissociation was exactly the active ice. (2) Carbon dioxide hydrate formation-dissociation route. The procedures of this route are almost the same with route (1), except that the methane was replaced by carbon dioxide. (3) ice powder or natural snow route. A certain amount of ice was ground into ice powder. Then, some ice powder or natural snow was mixed with surfactant evenly. The grinding and mixing are conducted under the protection of dry ice.

### Compress the active ice

In the experiments shown in Fig. 4d, e, the active ice was compressed before being used in hydrate formation. The active ice was cooled to 268.15 K first. Then, a precooled PTFE disk with similar diameter to the internal diameter of the sapphire reactor was put into the reactor and placed on the active ice. Subsequently, the piston was pushed upward by the hand pump, and the active ice, therefore, was compressed between the piston and the PTFE disk. Before the compression, the pressurizing fluid was precooled in the buffer tank to avoid the dissociation of the active ice. After the compression, the PTFE disk was taken out from the reactor, and 0.2 MPa of methane was charged into the reactor to push the piston back to the bottom of the reactor. Then, the reactor was discharged to atmospheric pressure. All these operations were conducted at 268.15 K.

### Methane hydrate formation in the active ice

The hydrate formation experiments in the active ice were mainly conducted in the sapphire reactor depicted in Supplementary Fig. 13; namely, the active ice was not taken out from the sapphire reactor after it was prepared. The temperature for hydrate formation was reset on the air bath. The gas reservoir was charged with methane to 10 MPa. 3 h later, the temperature of the active ice and methane gas reached the desired value. Then, the reactor was charged with methane to 6.0 MPa from the gas reservoir. The pressure drop in the reactor was used to calculate the gas uptake in the hydrate. In the control experiments that methane hydrate formation in SDS solution, the active ice was displaced by a certain amount of SDS solution; the following steps were the same with the hydrate formation in the active ice.

### Gas separation with the active ice

The gas separation experiments were also conducted in the apparatus depicted in Supplementary Fig. 13. After the active ice was prepared in the reactor, the temperature for hydrate formation was reset on the air bath. Gas mixture was injected into the gas reservoir to a certain pressure. When the temperature of the active ice and the gas mixture reached the set value, the reactor was pressurized with the gas mixture from the gas reservoir. When the pressure in the reactor stopped decreasing, the gas remained in the reactor was analyzed with an Agilent 7890 A gas chromatography.

## Data treatment

Both the gas storage and separation results were calculated based on the molar balance of gas in the gas and hydrate phase. The detailed treatment is shown in Supplementary Information.

## Characterization

The methods of Raman, PXRD, SEM, and DSC characterization are shown in Supplementary Information.

## Data availability

The data generated in this study are provided in the Supplementary Information and Source Data file. Source data are provided with this paper.

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

## Acknowledgements

The authors acknowledge fundings from the National Natural Science Foundation of China (Grant No. 22127812 to G.-J.C., 52004311 to P.X., U20B6005 to G.-J.C., and 22178379 to C.-Y.S.).

## Author contributions

G.-J.C., P.L., and B.L. designed the project; C.-Y.S. and P.X. designed the experiments. P.X. performed the active ice preparation and hydrate formation with the help of J.-J.L. and X.W.P.; P.X. carried out the analysis with the help of W.-X.P., X.-H.W., C.D., B.L., and Y.-L.P. Y.X. and Y.-J.Z. performed the Raman and DSC characterization. W.C. contributed to the gas separation experiments. P.X. drafted the manuscript. G.-J.C. and P.L. reviewed and edited the manuscript.

## Competing interests

The authors declare no competing interests.
