## [Peer Review File · Nature Communications]

REVIEWER COMMENTS

Reviewer #1 (Remarks to the Author):

Interesting work but offers more questions than answers.

How did the induction times, the gas uptake rates, the total conversion amounts, etc., compare to pure fine ice powers of various grain sizes? Figure 1 shows an example comparison at 6 MPa. What was the size distributions of the ice powder? Quite obviously, the finer the powder size the more conducive to the methane hydrate formation. This point aside, I suspect that the difference between them will shrink with the methane gas pressure.

The authors claim that "the active ice can be easily regenerated by depressurization below the ice point." How? Do the authors mean that the structure (the porous structure made by the dissociation of the SDS solution below the ice point or the grain size distributions of the ice powder) can be preserved over the ice-to-clathrate-to-ice conversion cycles despite the likely generation of quasi liquid layers (or the "aqueous solution layer", as the authors put it) in between the cycles? How is this possible? Conversely, if an irreversible time evolution of the structures takes place, what is the required initial condition that allows regeneration and how long can it last? In fact, the authors' explanations of possible mechanisms, with the schematic diagrams, underscores this point.

What happens to the regeneration capabilities after compression? Is it really possible to have your cake and eat it too?

Reviewer #2 (Remarks to the Author):

This study proposes the enhancement of gas hydrate kinetics and its applicability in gas separation processes using a unique seed called "active ice." After converting the SDS solution into hydrate, the resulting "active ice" obtained by melting the hydrate at temperatures below freezing point exhibited remarkable improvement in kinetics. However, it lacks distinctiveness compared to studies utilizing ice powder or hydrate seed, which have been extensively reported in the existing literature. To provide more meaningful research directions and enhance the quality of the paper based on the conducted experiments, it is necessary to incorporate additional information or make revisions to address the commented points.

The following are comments; it is essential to provide in-depth responses to comments 1 and 2.

1. Supplementary Fig.1 and Fig.15 illustrate the method of producing active ice and the temperature/pressure changes during its formation. The most important aspect of active ice formation is to rapidly depressurize the formed hydrate at temperatures below the freezing point, allowing only the hydrate to dissociate without melting the ice. However, the observed reactor temperature does not exhibit a significant decrease during depressurization (4 → 0 MPa), which should be expected due to the Joule-Thomson effect. Considering the provided reactor volume (61.90 cm³) and the volume of the content (5-45g), a depressurization of 4 MPa should result in a sharp temperature decrease. A similar example (Scientific Reports, 9(1), 1-11. <https://doi.org/10.1038/s41598-019-48832-8>) reported a substantial temperature drop during depressurization, and therefore, a clear response or a simple replication experiment is recommended to address this discrepancy.

2. In relation to Comment 2, a question arises that if the temperature drops significantly due to the Joule-Thomson effect, not only ice but also hydrate can remain intact without melting for an extended period. The "active ice" is thought to be a composite crystal structure where hydrate is mixed with ice. Particularly, when a significant amount of hydrate remains within the active ice, it

raises ambiguity in the identity of active ice and may obscure the core focus of the paper. This is because it diminishes the differentiation from experiments that utilize hydrate as a seed, and each time active ice is regenerated, the gas uptake decreases. Therefore, it is requested provide clearer verification, as presented in Supplementary Figure 15, that there is minimal remaining hydrate within the formed active ice after depressurization. This can be adequately confirmed by comparing the amount of dissociated gas during the recycling experiments or through real-time Raman analysis of the active ice itself.

3. Fig.1e and 1f on page 5 depict the kinetics of methane hydrate formation in SDS solution and active ice, respectively. While comparing these two cases is highly reasonable, it should be noted that the experimental conditions differ (the experiment in Fig. 1e is conducted at 277.15 K and 6.0 MPa, while Fig. 1f is at 272.65 K and 6.0 MPa). Due to the significant temperature difference ($\Delta T = 4.5\text{K}$), it is expected that the SDS solution case would naturally exhibit unfavorable aspects such as induction time or gas uptake. Is there a specific reason for conducting the experiments with different temperature conditions for Fig. 1e and 1f?

4. In this paper, the conversion rate of hydrates is presented as V/V_w . There is no doubt that V/V_w is one useful method to indicate the conversion rate of hydrates. However, the value of V/V_w can vary significantly depending on the hydrate structure, types of additives, and hydration number in the specific system. Therefore, to avoid confusion for readers, it is recommended to provide supplementary explanations regarding the conditions and methods used to calculate V/V_w and to include the theoretically maximum value for the corresponding system.

5. It is recommended to adjust the brightness and contrast of the SEM image in Fig. 2c (page 7) to enhance the clarity of the porous structure. The current image appears blurry, and it would be beneficial to show the porous structure more clearly by adjusting the brightness and contrast of the original image. Additionally, in the Supplementary section, it is advised to provide clear specifications of the experimental methods during SEM measurements, such as the temperature range and acceleration voltage.

6. To determine the crystalline phases of the samples in Fig 2a (page 7), it appears necessary to compare the diffraction pattern with that of ice in the reference, using markers such as asterisks, or employing the Rietveld refinement method.

7. The Raman spectrum depicted in Figure 2 on page 7 illustrates the variations in methane uptake before and after the activation of ice. However, there appears to be a lack of information concerning the small undefined peak observed at the 3050 cm^{-1} point. Additionally, Supplementary Figure 3 provides a simple comparison between subcooled water and pure ice, which seems unrelated to active ice.

8. It is recommended to set the y-axis scale consistently to facilitate comparison (in Fig. 4, Supplementary Fig. 7, etc.)

9. It is recommended to plot Fig. 5d (page 10 in Supplementary information) using color-coded lines instead of large symbols, as the current size of the symbols makes it difficult to discern the temperature path.

10. Fig. 1 in the Supplementary section contains important visual information about the concept of producing active ice. It is considered that combining this information with Fig. 5 (page 15) would enhance the visual representation and effectively convey the key findings of the research.

Reviewer #3 (Remarks to the Author):

Review of the manuscript "Prompt formation of hydrate from active ice with high gas uptake" written by Peng Xiao et al.

In the manuscript presented here, the authors describe the use of so-called active ice for the efficient formation of gas hydrates with a simultaneously very high gas uptake. To produce the activated ice, an aqueous solution is mixed with small doses of sodium dodecyl sulfate (SDS) and this is reacted with methane (or CO₂) to form hydrate. The resulting hydrate is decomposed at temperatures below the freezing point and the active ice is obtained. The approach appears promising for a number of applications. The manuscript is very well written and the results are well presented, nevertheless, a few points are at least unclear to me and I would like to suggest a few improvements that can contribute to the understanding of the subject.

Page 6, line 140: It is not clear to me why Figure 3 in the supplementary material is referred to at this point, since this figure shows nothing to indicate a transformation to hydrate, as described in the text.

Page 7, lines 153-158: The authors compare the effect of the SDS molecules with that of salt, resulting in a lowering of the freezing point. However, the measured melting points in supplemental Table 2 are higher than the melting point of pure ice (273.17-273.24 K versus 273.15 K for pure ice). Therefore, the argument that the presence of the SDS molecules leads to a liquid aqueous phase even at temperatures around 272.65 K is not really conclusive. It would also be interesting if the standard deviation of the measurements for the melting points were given. Page 9, Figure 3: Figure 3: the mechanism proposed here probably represents only part of a more complex conversion reaction. The porous structure of the activated ice will probably be saturated with the methane gas in the pores, which improves the availability of the gas throughout the sample. This would also explain why gas uptake decreases with compaction (= less pore space). It is also questionable whether the role of the SDS molecules is limited to forming a liquid water phase, or whether the SDS molecules do not also have an effect on the local orientation of the water molecules, which could favor hydrate formation.

Page 10, lines 210-215 and also supplementary Figure 5: The data for the sample using 79.0 g of activated ice are not presented in this Figure. Therefore, one cannot follow the data discussed here. Please add the missing data to supplementary Figure 5.

Page 13, Figure caption: It is very interesting that also snow mixed with SDS shows a similar effect. This means that at least no "memory effect" is necessary for the rapid hydrate formation. What I wonder, though, is whether experiments have also been done with compressed snow?

Page 20, lines 419-426: Since it is not explicitly mentioned here, I wonder if GC measurements were also made from the released gas phase after hydrate decomposition? Because these data would be needed for equation 6 in the Supplementary Material. You may add this information.

Supplementary Material:

Page 10, Figure 5a: Why does the temperature remain high (above 278 K) after the first transformation of active ice has been completed and the gas uptake has already been sharply reduced again? This also does not seem to be consistent with the temperature curve shown in figure 5b.

Also: please add data for the 79.0 g sample in Figure 5c (see comment above).

Page 13, Figure 7: Despite the remarkably high increase in gas uptake in the first 5 min, the curves shown in Figs. 7c-f in particular show a relatively "classical" progression, the initial steep slope indicating initial strong hydrate formation, whereas the flattening slope from about 10 min onward indicates a diffusion-controlled process. Perhaps this aspect should be considered in the manuscript.

Response to reviewers

Point-to-point response

Reviewer #1 (Remarks to the Author):

Interesting work but offers more questions than answers.

1. How did the induction times, the gas uptake rates, the total conversion amounts, etc., compare to pure fine ice powders of various grain sizes? Figure 1 shows an example comparison at 6 MPa. What was the size distributions of the ice powder? Quite obviously, the finer the powder size the more conducive to the methane hydrate formation. This point aside, I suspect that the difference between them will shrink with the methane gas pressure.

Reply: The particle size of the ice powder used in Fig. 1 was $180 \sim 250 \mu\text{m}$, because it was acquired by sieving ice powder with 60 and 80-mesh sieves. We measured the particle size distribution of the sieved ice powder by Focused Beam Reflectance Measurement (FBRM) at 263.15 K, and the chord length distribution is shown below (Fig. R1a). The chord length distribution represents the size distribution. Because the ice particles were not spherical, a small portion of the ice grains were less than $180 \mu\text{m}$ or larger than $250 \mu\text{m}$.

We further prepared some ice powder by finely grinding and then sieving with an 140-mesh sieve. Thus, the size of the newly prepared ice powder was smaller than $106 \mu\text{m}$. 140-mesh was the smallest size that the ice powder could pass through in our experiments. We used this ice powder for methane hydrate formation at 272.65 K and initial pressure of 6.0 MPa. The gas uptake profile is shown in Fig. R1b. In comparison with the ice powder of $180 \sim 250 \mu\text{m}$, the hydrate formation in finer ice powder was relative faster, just as you mentioned; however, it was still much slower than that in the active ice (Fig. 1f on page 5 in manuscript).

Therefore, the gas uptake rate and ice conversion are much higher when gas hydrate is formed in the active ice.

As for the induction times of gas hydrate formation in the active ice and in the finer ice powder, they were very close. We noticed that in some experiments of gas hydrate formation in the active ice, once the gas pressure slightly exceeded the equilibrium value (i.e., 1 MPa) during charging gas into the reactor, the pressure dropped rapidly, and sometimes the charging rate of gas even couldn't catch up with its consuming rate. When methane hydrate formed in fine ice powder, gas hydrate formation was observed after the gas injection at once.

Unquestionably, the hydrate formation would be faster in ice powder with smaller particle size or under higher driving force. However, this perhaps mainly occurs at the beginning of the hydrate formation. As shown in Fig. R1b, methane hydrate formed much faster in the ice powder with smaller grains; however, this fast formation stage only existed for 6 minutes and the gas uptake was only 15.43 V/V_w before the formation slowed down; after that, the slopes of the two gas uptake curves changed to similar, which suggested that the formation rates were similar and the formation regime went into a mass transfer-controlled stage. It can be inferred from the comparison that once the size of the ice particle is small enough, like less

than twice the initial thickness of hydrate film, the gas hydrate formation could be finished within a short time. However, according to the initial thickness measurements by Li et al. (AIChE J 59, 2145-2154 (2013)), the particle size should be less than 13.2~39.62 μm at driving force of 1.0 ~ 3.0 K for methane hydrate formation, and ice powder with such size is difficult to produce in large scale.

Fig. R1. Size distribution of the pure ice powder used in the manuscript (a), and methane uptake in the pure ice powder with particle size of 180 ~ 250 μm and less than 106 μm (b).

2. The authors claim that “the active ice can be easily regenerated by depressurization below the ice point.” How? Do the authors mean that the structure (the porous structure made by the dissociation of the SDS solution below the ice point or the grain size distributions of the ice powder) can be preserved over the ice-to-clathrate-to-ice conversion cycles despite the likely generation of quasi liquid layers (or the “aqueous solution layer”, as the authors put it) in between the cycles? How is this possible? Conversely, if an irreversible time evolution of the structures takes place, what is the required initial condition that allows regeneration and how long can it last? In fact, the authors’ explanations of possible mechanisms, with the schematic diagrams, underscores this point.

Reply: We thank the reviewer for this insightful comment and question. Actually, the regeneration of the active ice does not refer to strictly reinstating the microstructure of the active ice, it only means recovering the ability of providing ultra-fast gas uptake as evidenced in our work.

The active ice was regenerated by depressurizing the gas hydrate to atmospheric pressure at temperature below the ice point in this work. The performance of recycled active ice in gas hydrate formation was still closely related to the porous structure of ice and the distribution of kinetic promoter, just as the fresh active ice. Definitely, the active ice particle would change its shape after experiencing gas hydrate formation and dissociation — one particle could change from an irregular porous shape to another one, or particles connected to each other. However, the porous characteristic is still preserved during gas hydrate formation and dissociation. This is because: (1) though most of the active ice melt within 5 minutes according to the formation time, no liquid water accumulates because the melt water converts to gas hydrate immediately once it appears; similarly, liquid water does not accumulate during slow dissociation of gas hydrate conducted below the ice point; thus, no large amount of

liquid water exists in the active ice or gas hydrate within a short time range, which helps to prevent the porous active ice or ice particles from converting to ice block; (2) gas hydrate formation in SDS solution create porous structure as we know, and gas hydrate dissociation below ice point also produce pores in the produced ice (Falenty A. et al., Energy Fuels 28, 6275-6283 (2014)). Therefore, the porous structure of the active ice could be preserved during gas hydrate formation and dissociation.

In summary, though irreversible structure change would inevitably happen to the active ice particles during hydrate formation and dissociation, the active ice is still able to retain its porous structure and is able to reproduce ultra-fast gas uptakes with high storage capacity in repeat cycles.

3. What happens to the regeneration capabilities after compression? Is it really possible to have your cake and eat it too?

Reply: We measured the methane uptake of compressed active ice at temperature of 272.65 K and initial pressure of 6.0 MPa. The active ice was prepared by one of the methods introduced in the manuscript: producing ice powder by grading, and directly adding SDS into the ice powder. The size of ice particle was 180 ~ 250 μm . The loosening coefficient was 1.252.

The methane uptake rate is shown in Fig. R2. As shown, the hydrate formation rate in the compressed active ice was as fast as that in the uncompressed active ice presented in the manuscript. Therefore, with a moderate compression, the regeneration of the active ice is satisfactory.

However, we also found that excessive compression (like loosening coefficient smaller than 1.1) would result in some problems. Apart from the gas uptake loss shown in the manuscript, it was found that the compressed active ice column would burst into pieces during hydrate dissociation, which are unfavorable for practical use.

Therefore, decreasing the apparent volume of the active ice (or gas hydrate) by compression and good regeneration capacity can be realized at the same time. However, this only happens at moderate compression levels.

Fig. R2. Comparison of methane uptake in 600-ppm SDS solution and in compressed active

ice at 272.65 K and initial pressure of 6.0 MPa.

Reviewer #2 (Remarks to the Author):

This study proposes the enhancement of gas hydrate kinetics and its applicability in gas separation processes using a unique seed called "active ice." After converting the SDS solution into hydrate, the resulting "active ice" obtained by melting the hydrate at temperatures below freezing point exhibited remarkable improvement in kinetics. However, it lacks distinctiveness compared to studies utilizing ice powder or hydrate seed, which have been extensively reported in the existing literature. To provide more meaningful research directions and enhance the quality of the paper based on the conducted experiments, it is necessary to incorporate additional information or make revisions to address the commented points.

The following are comments; it is essential to provide in-depth responses to comments 1 and 2.

1. Supplementary Fig.1 and Fig.15 illustrate the method of producing active ice and the temperature/pressure changes during its formation. The most important aspect of active ice formation is to rapidly depressurize the formed hydrate at temperatures below the freezing point, allowing only the hydrate to dissociate without melting the ice. However, the observed reactor temperature does not exhibit a significant decrease during depressurization (4 → 0 MPa), which should be expected due to the Joule-Thomson effect. Considering the provided reactor volume (61.90 cm³) and the volume of the content (5-45g), a depressurization of 4 MPa should result in a sharp temperature decrease. A similar example (Scientific Reports, 9(1), 1-11. <https://doi.org/10.1038/s41598-019-48832-8>) reported a substantial temperature drop during depressurization, and therefore, a clear response or a simple replication experiment is recommended to address this discrepancy.

Reply: The Joule-Thomson effect occurred during depressurization in fact, but the temperature sensor was covered by gas hydrate, it therefore was not detected. In that experiment, we stirred the SDS solution to accelerate gas hydrate formation, and some hydrate attached on and gradually covered the temperature sensor finally though the stirring device was switched off after hydrate formation began.

We conducted two experiments to demonstrate if the cover of gas hydrate on temperature sensor could hinder the detection of Joule-Thomson effect. In one experiment, the temperature sensor was buried in gas hydrate formed from 10 g active ice, and in another one, the reactor was empty so the sensor was completely exposed in gas. As shown in Fig. R3, when the reactor was depressurized to atmospheric pressure from 4.0 MPa suddenly, the sensor detected a temperature decrease of 0.87 K when it was buried in gas hydrate, which was consistent with that in Fig. 15 (now it's Fig. 16) in the supplementary material. However, a rapid temperature decrease of 11.29 K was detected when the sensor was exposed in methane. This suggests that the Joule-Thomson effect was also strong in our experiments, and

it was not detected only because the temperature sensor was covered by hydrate. We have pointed it out in the caption of Fig. 16 (page 21 in the revised supplementary material).

Fig. R3. Comparison of temperature decrease during depressurization when temperature sensor was buried in hydrate and exposed in gas.

2. In relation to Comment 2, a question arises that if the temperature drops significantly due to the Joule-Thomson effect, not only ice but also hydrate can remain intact without melting for an extended period. The "active ice" is thought to be a composite crystal structure where hydrate is mixed with ice. Particularly, when a significant amount of hydrate remains within the active ice, it raises ambiguity in the identity of active ice and may obscure the core focus of the paper. This is because it diminishes the differentiation from experiments that utilize hydrate as a seed, and each time active ice is regenerated, the gas uptake decreases. Therefore, it is requested provide clearer verification, as presented in Supplementary Figure 15, that there is minimal remaining hydrate within the formed active ice after depressurization. This can be adequately confirmed by comparing the amount of dissociated gas during the recycling experiments or through real-time Raman analysis of the active ice itself.

Reply: We conducted a hydrate dissociation experiment to demonstrate that there was no residual hydrate existed in the active ice before it was used for gas hydrate formation in our work. Methane hydrate formation in 10.0 g active ice was performed at 272.65 K and initial pressure 6.0 MPa. Then the reactor was depressurized to atmospheric pressure to dissociate the hydrate at 272.65 K. Once the pressure reached to 1 atm, the reactor was connected to a water displacement device. 3.5 hours later, the temperature of the reactor was increased to 293.15 K to melt all the solid in the reactor. The methane uptake rate and gas release rate (indicated by the water displacement rate) are shown in Fig. R4.

As shown in Fig. R4a, the methane uptake was 190.9 V/Vw, namely, 1909 cm³ STP methane was stored in the hydrate. From Fig. R4b, it could be found that no more water was displaced after hydrate dissociated for 1.5 hour, and 1917 g water was displaced by released gas. Because the gas temperature in the water displacement device was 297.65 K, the volume of gas released should be 1759 cm³ at standard temperature and pressure. When heating the reactor to 293.15 K, only 15.17 g water was replaced by released gas, which was mainly caused by the volume expansion due to temperature rise. This suggested that hydrate dissociation finished before 1.5 h, and the difference between gas fixed in hydrate (1909 cm³)

and gas collected during dissociation (1759 cm^3) was because the fast dissociation during discharging the reactor.

Therefore, after a long enough dissociation period (like 3 hours in the manuscript) at 272.65 K and 1 atm , no gas hydrate was present in the active ice. Another point of view is that, if there would have been some undissociated gas hydrate present with active ice, in the repeat cycles, we would have detected a decrease in the gas uptake which we did not observe. This reaffirms that there was no residual hydrate present in the active ice when it was subjected to hydrate formation. In addition, we have confirmed that by mixing SDS with pure ice powder or natural snow, the mixture also showed excellent performance in gas hydrate formation (Fig. 4f on page 13 in the revised manuscript). Compared with active ice prepared from fresh ice, the active ice prepared by dissociating gas hydrate showed no advantage in hydrate formation, which suggests that even if some gas hydrate survives from the dissociation and remains in the active ice, it has no obvious positive effect on gas hydrate formation.

Fig. R4. Methane uptake rate when hydrate formed in 10.0 g active ice (a), and mass of water displaced during gas hydrate dissociation at 272.65 K and 1 atm (b).

3. Fig. 1e and 1f on page 5 depict the kinetics of methane hydrate formation in SDS solution and active ice, respectively. While comparing these two cases is highly reasonable, it should be noted that the experimental conditions differ (the experiment in Fig. 1e is conducted at 277.15 K and 6.0 MPa , while Fig. 1f is at 272.65 K and 6.0 MPa). Due to the significant temperature difference ($\Delta T = 4.5 \text{ K}$), it is expected that the SDS solution case would naturally exhibit unfavorable aspects such as induction time or gas uptake. Is there a specific reason for conducting the experiments with different temperature conditions for Fig. 1e and 1f?

Reply: This is a very good comment by the reviewer and we thank him for highlighting this. Our original concern was that of the freezing problem, because before gas injection, the solution may freeze if the formation experiment is conducted at temperature lower than ice point.

However, upon the reviewer's comment, we performed a few experiments on forming gas hydrate from SDS solution at such low temperature, and the results showed no freezing problem. Therefore, to make a direct comparison between the gas hydrate formation in SDS solution and in the active ice at the same temperature, we conducted three experiments of

methane hydrate formation in 600-ppm SDS solution at 272.65 K and initial pressure of 6.0 MPa. Hydrate formation was accelerated by stirring, and the stirring device was switched off at once when hydrate appeared in the reactor.

Fig. R5 shows the methane uptake profiles. As shown, the induction time of hydrate formation in SDS solution was stochastic even when stirring was adopted. From our experience, the induction time in a sapphire reactor is much remarkable than that in a stainless steel reactor, perhaps because the sapphire reactor is smoother. In addition, the time required to finish gas hydrate formation since it started was 21, 18 and 32 mins for run 1, 2 and 3, respectively. For comparison, in all the hydrate formation experiments in the active ice, hydrate nucleation was very fast and the hydrate formation rate was ultra-fast.

Therefore, the gas hydrate formation in SDS solution at the same temperature was also much slower than that in the active ice. To avoid the readers having the same question, the methane uptake profile at 277.15 K in Fig. 1e (page 5 in the revised manuscript) has been replaced by that at 272.65 K (run 1 in Fig. R5). The Fig. R5 has been edited and put into the supplementary material as Fig. 2 on page 7, and the corresponding explanation has been added on page 4 in the manuscript.

Fig. R5. Methane uptake in 600-ppm SDS solution at 272.65 K and initial pressure of 6.0 MPa.

4. In this paper, the conversion rate of hydrates is presented as V/V_w . There is no doubt that V/V_w is one useful method to indicate the conversion rate of hydrates. However, the value of V/V_w can vary significantly depending on the hydrate structure, types of additives, and hydration number in the specific system. Therefore, to avoid confusion for readers, it is recommended to provide supplementary explanations regarding the conditions and methods used to calculate V/V_w and to include the theoretically maximum value for the corresponding system.

Reply: We provided the method to calculate V/V_w in Section I in Supplementary material, however, the important information that you mentioned was missed out. As you know, methane forms structure I hydrate at the temperature and pressure that we tested, and the ideal formula of structure I hydrate is $8M \cdot 46H_2O$. For the ideal situation that all the cages are occupied by methane molecules, the ideal value of V/V_w would be 216. The additives used in

this study are surfactants, which are known to do not affect ideal storage capacity of gas hydrate because they do not occupy cages of gas hydrate. We have added these explanations in the revised supplementary material (page 4~5 in manuscript, page 3 in supplementary material).

5. It is recommended to adjust the brightness and contrast of the SEM image in Fig. 2c (page 7) to enhance the clarity of the porous structure. The current image appears blurry, and it would be beneficial to show the porous structure more clearly by adjusting the brightness and contrast of the original image. Additionally, in the Supplementary section, it is advised to provide clear specifications of the experimental methods during SEM measurements, such as the temperature range and acceleration voltage.

Reply: We have adjusted the brightness and contrast of the SEM image in Fig. 2c, Fig. 2b in manuscript (page 7) and Fig. 3 in supplementary material (page 8), and the structure seems clearer now. A more detailed procedure of SEM imaging has been added to the supplementary material (page 5), and both the temperature of cryo-stage and the acceleration voltage have been involved.

6. To determine the crystalline phases of the samples in Fig 2a (page 7), it appears necessary to compare the diffraction pattern with that of ice in the reference, using markers such as asterisks, or employing the Rietveld refinement method.

Reply: We have added the XRD pattern of ice Ih provided by Bernal et al.'s work (Bernal JD, Fowler RH. *The Journal of Chemical Physics* 1, 515-548 (1933).) to Fig. 2a (page 7 in the revised manuscript). The asterisk refers to the main peaks.

7. The Raman spectrum depicted in Figure 2 on page 7 illustrates the variations in methane uptake before and after the activation of ice. However, there appears to be a lack of information concerning the small undefined peak observed at the 3050 cm^{-1} point. Additionally, Supplementary Figure 3 provides a simple comparison between subcooled water and pure ice, which seems unrelated to active ice.

Reply: The peak at 3050 cm^{-1} belongs to the O-H stretching in structure I methane hydrate. Schicks et al. indicates that the O-H stretching region locates at 3000~3800 cm^{-1} (Schicks JM, Erzinger J, Ziemann MA. *Spectrochimica Acta Part a-Molecular and Biomolecular Spectroscopy* 61, 2399-2403 (2005).). A big difference between ice Ih and structure I methane hydrate in O-H stretching is if there exists a peak at 3050 cm^{-1} , however, to our best knowledge, what the small peak at 3050 cm^{-1} exactly indicates haven't been studied yet. Because the peak at 2904 and 2914 cm^{-1} is sufficient to indicate the formation of methane hydrate in the active ice, we therefore ignored the peak at 3050 cm^{-1} . We have pointed out in the caption of Fig. 2 (on page 7 in the revised manuscript), that the peak at 3050 (3053 in fact) cm^{-1} is the distinction between O-H stretching in methane hydrate and ice.

The purpose of Fig. 3 in supplementary material is to compare the Raman spectra of methane hydrate, the active ice, pure ice, and subcooled water. It seems that the spectra of subcooled water indeed had no use because we can distinguish ice and water by eyes, so we delete it. In

addition, we put the spectrum of pure ice into Fig. 2d on page 7 in the revised manuscript for comparison, and the Fig. 3 in the supplementary material has been completely deleted.

8. It is recommended to set the y-axis scale consistently to facilitate comparison (in Fig. 4, Supplementary Fig. 7, etc.)

Reply: We have reset the y-axis scale consistently in the figures that include gas uptake profiles like Figs. 1, 4 on pages 5 and 13 in the manuscript, and Figs 6, 7, 8 on pages 11-13 in the supplementary material.

9. It is recommended to plot Fig. 5d (page 10 in Supplementary information) using color-coded lines instead of large symbols, as the current size of the symbols makes it difficult to discern the temperature path.

Reply: We have re-plot the Fig. 5d as mentioned using color-coded lines. In addition, we found that we missed out on the data of hydrate formation in 79.0 g active ice in Fig. 5c and Fig. 5d on page 10 in the supplementary material, we have revised these two figures and included the missing data.

10. Fig. 1 in the Supplementary section contains important visual information about the concept of producing active ice. It is considered that combining this information with Fig. 5 (page 15) would enhance the visual representation and effectively convey the key findings of the research.

Reply: We have reorganized the Fig. 5 (page 16 in the revised manuscript), and the main information of Fig. 1 supplementary material has been included.

Reviewer #3 (Remarks to the Author):

Review of the manuscript "Prompt formation of hydrate from active ice with high gas uptake" written by Peng Xiao et al.

In the manuscript presented here, the authors describe the use of so-called active ice for the efficient formation of gas hydrates with a simultaneously very high gas uptake. To produce the activated ice, an aqueous solution is mixed with small doses of sodium dodecyl sulfate (SDS) and this is reacted with methane (or CO₂) to form hydrate. The resulting hydrate is decomposed at temperatures below the freezing point and the active ice is obtained. The approach appears promising for a number of applications. The manuscript is very well written and the results are well presented, nevertheless, a few points are at least unclear to me and I would like to suggest a few improvements that can contribute to the understanding of the subject.

1. Page 6, line 140: It is not clear to me why Figure 3 in the supplementary material is referred to at this point, since this figure shows nothing to indicate a transformation to hydrate, as described in the text.

Reply: Figure 3 in the supplementary material was used to compare the Raman spectra of methane hydrate, the active ice, pure ice, and subcooled water. It seems that the spectrum of subcooled water indeed had no use because we can distinguish ice and water by eyes, so we delete it. In addition, we put the spectrum of pure ice into Fig. 2d (page 7 in revised manuscript) for comparison, and the Fig. 3 has been completely deleted from the supplementary material.

2. Page 7, lines 153-158: The authors compare the effect of the SDS molecules with that of salt, resulting in a lowering of the freezing point. However, the measured melting points in supplemental Table 2 are higher than the melting point of pure ice (273.17-273.24 K versus 273.15 K for pure ice). Therefore, the argument that the presence of the SDS molecules leads to a liquid aqueous phase even at temperatures around 272.65 K is not really conclusive. It would also be interesting if the standard deviation of the measurements for the melting points were given.

Reply: The conclusion that SDS leads to a liquid aqueous phase at 272.65 K was based on the latent heat when heating the SDS solutions from 253.15 K to 278.15 K, rather than the melting point. The melting heat reduces in the presence of SDS compared to pure ice (e.g., 1.14×10^4 mJ at SDS concentration of 5.09wt% versus 1.24×10^4 mJ in pure water with the same mass). This was more conspicuous at a higher SDS concentration. Though the concentration of SDS was only 600 ppm overall in the active ice, the local concentration in the small spaces that surround SDS molecules or SDS clusters could be high enough to prevent water from freezing.

From the latent heat in Table 2 in the supplementary material, it can also be found that some water kept liquid phase even at 253.15 K. According to that, we think that the melting points measured belonged to the ice parts excluding the unfrozen water. Though the local high concentration of SDS may leads to more local ice melt during heating, the endotherm of local ice melt had hardly any effect on the overall melting point measurement, otherwise, the measured melting points would be lower than 273.15 K. Therefore, we think that the melting point has barely any relation to the very little unfrozen water, and the deviation of the melting points came from measurement error of the DSC, which is ± 0.2 K as described in *Characterization* section (page 4) in the supplementary text.

3. Page 9, Figure 3: the mechanism proposed here probably represents only part of a more complex conversion reaction. The porous structure of the activated ice will probably be saturated with the methane gas in the pores, which improves the availability of the gas throughout the sample. This would also explain why gas uptake decreases with compaction (= less pore space). It is also questionable whether the role of the SDS molecules is limited to forming a liquid water phase, or whether the SDS molecules do not also have an effect on the local orientation of the water molecules, which could favor hydrate formation.

Reply: The mechanism of fast gas hydrate formation in the active ice is complex, just as you refer to. We agree with that the pores on ice particles and the gaps between the particles ensure the gas transfer inside the active ice bed, that's why gas hydrate formation was slow in over compressed active ice in our experiments.

SDS indeed caused liquid water in the active ice, because the melting heat of the ice which contains SDS is smaller than that of equivalent pure ice (we have explained this in the reply to the previous question). Many studies on gas hydrate formation in SDS solution suggest that gas hydrate grows mainly toward the space above the solution, then the water migrates along the formed hydrate and finally forms porous hydrate. Therefore, we can infer that the melt water also migrates towards the space around the ice particle, and SDS is responsible for such migration. According to this, we have revised Fig. 3 on page 9 in the revised manuscript.

4. Page 10, lines 210-215 and also supplementary Figure 5: The data for the sample using 79.0 g of activated ice are not presented in this Figure. Therefore, one cannot follow the data discussed here. Please add the missing data to supplementary Figure 5.

Reply: We are sorry that we omitted the data of 79.0 g in the figure, and thanks for pointing it out. We have added the missing data to the figure (Figure 5 on page 10 in the revised supplementary material).

5. Page 13, Figure caption: It is very interesting that also snow mixed with SDS shows a similar effect. This means that at least no "memory effect" is necessary for the rapid hydrate formation. What I wonder, though, is whether experiments have also been done with compressed snow?

Reply: According to our experiment record, the snow was slightly compressed to the volume of 1.5 times equivalent ice block.

6. Page 20, lines 419-426: Since it is not explicitly mentioned here, I wonder if GC measurements were also made from the released gas phase after hydrate decomposition? Because these data would be needed for equation 6 in the Supplementary Material. You may add this information.

Reply: The GC measurements were only conducted for the gas phase reminded in the reactor and the feed gas. As for the composition of the gas stored in hydrate, analyzing the released gas by GC could not acquire accurate composition because some hydrate would dissociate when discharging the reactor to atmospheric pressure. It was acquired by calculating.

We have checked the program that we used for data treatment, which was modified from the data processing method in our previous work (Chem Eng J 336, 649-658 (2018).), and found that the method described in the data treatment section in supplementary information do not match the method that we actually use (the method in supplementary information simplified the actual method used to treat the data). We are sorry for the confusion caused by our mistake. We have corrected the data treatment section in the revised supplementary material.

The gas composition in the hydrate was acquired by calculating.

The equation 11 and 12 on page 3 in the supplementary material are listed below,

$$N_{in} z_1 = N_{gas} y_1 + N_{hydrate} x_1$$

$$N_{in} z_2 = N_{gas} y_2 + N_{hydrate} x_2$$

where N_{in} is the number of moles of gas injected into the reactor. N_{gas} is the number of moles

of gas remained in the reactor. $N_{hydrate}$ refers to the number of moles of gas stored in gas hydrate. z_1 and z_2 are the molar fractions of component 1 and 2 in the feed gas. y_1 and y_2 are the molar fractions in the remained gas in the reactor. z_1 , z_2 , y_1 and y_2 were measured by GC. N_{in} and N_{gas} can be calculated by EoS equation.

$$N_{in} = \frac{P_{a,0}V_a}{Z_{a,0}RT} - \frac{P_{a,t}V_a}{Z_{a,t}RT}$$

When calculate N_{gas} , because the volume of gas is related to the volume of hydrate and residual ice (the gas volume kept changing during hydrate formation), we first calculate the ratio of ice converted to hydrate according to equation 6, which is pasted below

$$x_t = \left[N_{in,t} - \frac{P_{gas,t}(V_b - V_{stirrer} - 1.087V_w)}{Z_{gas,t}RT} \right] / \left(\frac{m_w}{108} - \frac{0.163P_{gas,t}V_w}{Z_{gas,t}RT} \right)$$

then we calculate the $N_{hydrate}$ according to equation 4, which is

$$N_{hydrate,t} = \frac{m_w x_t}{18 \times 6}$$

Then, only N_{gas} , x_1 and x_2 are unknown in equations 12 and 13. In combination with equation 1 on page 2 in the supplementary material, which is as below, N_{gas} can be acquired.

$$N_{in} = N_{gas} + N_{hydrate}$$

Finally, only x_1 and x_2 are unknown in equations 12 and 13. We can acquire them by solving these two equations and use them to calculate the separation factor in equation 10 (the equation 6 you mentioned).

That's how we calculated the gas composition in hydrate. We have added the information in the data treatment section to make it clear to readers.

Supplementary Material:

7. Page 10, Figure 5a: Why does the temperature remain high (above 278 K) after the first transformation of active ice has been completed and the gas uptake has already been sharply reduced again? This also does not seem to be consistent with the temperature curve shown in figure 5b.

Also: please add data for the 79.0 g sample in Figure 5c (see comment above).

Reply: Perhaps, we didn't make the figure clear for readers, or the reviewer misunderstand the figure, because the curves correspond to the axis of the same color. The Fig. 5a and 5b on page 10 in the supplementary material are the figures about the gas uptake and temperature change (Fig. 4a) and temperature–pressure change (Fig. 4b) during methane hydrate formation in 10.0 g 600-ppm SDS aqueous solution. In Fig. 4a, the gas uptake stopped at 32 min after a rapid increase; the temperature increased rapidly at first then reduced sharply.

Here we modify the Fig. 4a and 4b you mentioned and paste them below (Fig. R6). The temperature reached 279.09 K at row 298 in Fig. R6b. Because the data was acquired every 5 seconds, thus the temperature reached 279.09 K at $298 \times 5 / 60 = 24.8$ min after gas injection in the experiment, which was consistent with the time that temperature reached the peak in Fig. R6a.

In addition, we have added the data of hydrate formation in 79.0 g active ice in Fig. 5c and 5d (page 10) in the revised supplementary material.

Fig. R6. Gas uptake and temperature change (a) and temperature–pressure change (b) during methane hydrate formation in 10.0 g 600-ppm SDS aqueous solution.

8. Page 13, Figure 7: Despite the remarkably high increase in gas uptake in the first 5 min, the curves shown in Figs. 7c-f in particular show a relatively "classical" progression, the initial steep slope indicating initial strong hydrate formation, whereas the flattening slope from about 10 min onward indicates a diffusion-controlled process. Perhaps this aspect should be considered in the manuscript.

Reply: The diffusion-controlled process mainly originated from the nature of gas hydrate formation in the presence of kinetic promoters. Though the active ice was regarded as the medium for gas hydrate formation, the formation essentially occurred in kinetic promoter solution as the mechanism we described in the manuscript. We have added a figure depicting methane uptake in different kinetic promoter solutions in supplementary material (new Figure 8 on page 13). By comparing it with Figure 6 (the Figure 7 you mentioned), it could be found that the gas uptake curve in active ice is similar to that in aqueous solution when the same kinetic promoter was used. Namely, if one kinetic promoter accelerates gas hydrate formation well in aqueous solution, it is also possible to accelerate gas hydrate formation in the active ice.

This suggests that besides leading to liquid water phase in the active ice, the ability of kinetic promoter in promoting gas hydrate formation in aqueous solution is also an important factor affecting its performance in active ice. We have added this explanation to the revised manuscript (page 9).

REVIEWERS' COMMENTS

Reviewer #1 (Remarks to the Author):

The authors provided satisfactory responses to my earlier comments. They just need to explicitly state these points to their manuscript as these would be most obvious questions a reader would have;

"Actually, the regeneration of the active ice does not refer to strictly reinstating the microstructure of the active ice, it only means recovering the ability of providing ultra-fast gas uptake as evidenced in our work."

"Therefore, decreasing the apparent volume of the active ice (or gas hydrate) by compression and good regeneration capacity can be realized at the same time. However, this only happens at moderate compression levels."

Reviewer #2 (Remarks to the Author):

This study proposes the enhancement of gas hydrate kinetics and its applicability in gas separation processes using a unique seed called "active ice." After converting the SDS solution into hydrate, the resulting "active ice" obtained by melting the hydrate at temperatures below freezing point exhibited remarkable improvement in kinetics. The questions and revision requests from the reviewer, based on the conducted experiments, were aimed at improving the quality of the paper and providing more meaningful research content. The additional experiments, explanations, and manuscript revisions carried out by the authors in response were satisfactory.

Reviewer #3 (Remarks to the Author):

Review of the revised manuscript Ultra-fast formation of hydrate from active ice with high gas uptake.

The authors have comprehensively addressed the criticisms and suggestions and have essentially implemented them. I have only minor comments to make.

1) The authors added the information: "Raman spectra for pure ice and the active ice before and after methane uptake; the peaks at 2904 cm^{-1} and 2914 cm^{-1} indicate the C-H stretching vibration of methane molecule in large and small cages of structure I methane hydrate, respectively; 3000 ~ 3600 cm^{-1} corresponds to O-H stretching; the peak at 3053 cm^{-1} is the distinction between O-H stretching in methane hydrate and ice". I think this is a misinterpretation of the cited work published by Schicks et al. The broad Raman band between 3000-3600 can be assigned to the O-H stretching, however, the Raman band at 3053 cm^{-1} should be assigned to C-H stretching of CH₄ (3020 cm^{-1} in the gas phase).

2) Title: In my opinion, the term "ultrafast" sounds a bit popular-scientific, so I would suggest "enhanced".

3) Different units are used in the manuscript (J/g versus kJ/mol). It would be nice if this could be unified.

4) Please add "cumulative" to gas uptake in the figure caption of Figure 4 to avoid any misunderstandings.

Response to reviewers

General response:

We sincerely thank the editor and all reviewers for their valuable feedback again. The reviewer comments are laid out below in black. Our response is given in blue, and the corresponding revision is shown in red in the revised manuscript.

Point-to-point response

Reviewer #1 (Remarks to the Author):

The authors provided satisfactory responses to my earlier comments. They just need to explicitly state these points to their manuscript as these would be most obvious questions a reader would have; "Actually, the regeneration of the active ice does not refer to strictly reinstating the microstructure of the active ice, it only means recovering the ability of providing ultra-fast gas uptake as evidenced in our work." "Therefore, decreasing the apparent volume of the active ice (or gas hydrate) by compression and good regeneration capacity can be realized at the same time. However, this only happens at moderate compression levels."

Reply: Thanks for the reviewer's kindly suggestion, the information that the reviewer mentioned would eliminate the confusions about the regeneration of the active ice for the readers. We have added the information in the revised manuscript (on page 8 and 9).

Reviewer #2 (Remarks to the Author):

This study proposes the enhancement of gas hydrate kinetics and its applicability in gas separation processes using a unique seed called "active ice." After converting the SDS solution into hydrate, the resulting "active ice" obtained by melting the hydrate at temperatures below freezing point exhibited remarkable improvement in kinetics. The questions and revision requests from the reviewer, based on the conducted experiments, were aimed at improving the quality of the paper and providing more meaningful research content. The additional experiments, explanations, and manuscript revisions carried out by the authors in response were satisfactory.

Reply: Thanks for the reviewer's kindly comment.

1. The Raman spectrum depicted in Figure 2 on page 7 illustrates the variations in methane uptake before and after the activation of ice. However, there appears to be a lack of information concerning the small undefined peak observed at the 3050 cm^{-1} point.

Additionally, Supplementary Figure 3 provides a simple comparison between subcooled water and pure ice, which seems unrelated to active ice.

Reply: The peak at 3050 cm^{-1} belongs to the O-H stretching in structure I methane hydrate. Schicks et al. indicates that the O-H stretching region locates at $3000\sim 3800\text{ cm}^{-1}$ (Schicks JM, Erzinger J, Ziemann MA. *Spectrochimica Acta Part a-Molecular and Biomolecular Spectroscopy* 61, 2399-2403 (2005).). A big difference between ice Ih and structure I methane hydrate in O-H stretching is if there exists a peak at 3050 cm^{-1} , however, to our best knowledge, what the small peak at 3050 cm^{-1} exactly indicates haven't been studied yet. Because the peak at 2904 and 2914 cm^{-1} is sufficient to indicate the formation of methane hydrate in the active ice, we therefore ignored the peak at 3050 cm^{-1} . We have pointed out in the caption of Fig. 2 (on page 7 in the revised manuscript), that the peak at 3050 (3053 in fact) cm^{-1} is the distinction between O-H stretching in methane hydrate and ice.

The purpose of Fig. 3 in supplementary material is to compare the Raman spectra of methane hydrate, the active ice, pure ice, and subcooled water. It seems that the spectra of subcooled water indeed had no use because we can distinguish ice and water by eyes, so we delete it. In addition, we put the spectrum of pure ice into Fig. 2d on page 7 in the revised manuscript for comparison, and the Fig. 3 in the supplementary material has been completely deleted.

2. I generally accept the author's explanations and responses. However, the reviewer recommends that the authors refer to the following paper: *J. Phys. Chem. B* 2009, 113, 15, 5172-5180.

Reply: We have carefully studied the paper that the reviewer mentioned, and cited it in the revised manuscript (reference 24).

Reviewer #3 (Remarks to the Author):

Review of the revised manuscript Ultra-fast formation of hydrate from active ice with high gas uptake. The authors have comprehensively addressed the criticisms and suggestions and have essentially implemented them. I have only minor comments to make.

1. The authors added the information: "Raman spectra for pure ice and the active ice before and after methane uptake; the peaks at 2904 cm^{-1} and 2914 cm^{-1} indicate the C-H stretching vibration of methane molecule in large and small cages of structure I methane hydrate, respectively; $3000\sim 3600\text{ cm}^{-1}$ corresponds to O-H stretching; the peak at 3053 cm^{-1} is the distinction between O-H stretching in methane hydrate and ice". I think this is a

misinterpretation of the cited work published by Schicks et al. The broad Raman band between 3000-3600 can be assigned to the O-H stretching, however, the Raman band at 3053 cm^{-1} should be assigned to C-H stretching of CH_4 (3020 cm^{-1} in the gas phase).

Reply: Thanks for pointing out this mistake, we have corrected in the revised caption of Fig. 2.

2. Title: In my opinion, the term "ultrafast" sounds a bit popular-scientific, so I would suggest "enhanced".

Reply: The reviewer raised a very professional suggestion. The slow formation rate of gas hydrate is the main cause hindering its practical application, and the most remarkable feature of this work is the breakthrough of the formation rate. Therefore, we request to keep a "ultra-rapid" word in the title to underline the breakthrough though the "ultra-rapid" also sounds a bit popular-scientific.

3. Different units are used in the manuscript (J/g versus kJ/mol). It would be nice if this could be unified.

Reply: The units of enthalpy have been unified into kJ mol^{-1} in the manuscript and the supplementary information.

4. Please add "cumulative" to gas uptake in the figure caption of Figure 4 to avoid any misunderstandings.

Reply: We have added this word to the caption of Figure 4.